# Vitamin D Deficiency Is a Potential Risk for Blood Pressure Elevation and the Development of Hypertension

**DOI:** 10.3390/medicina57121297

**Published:** 2021-11-25

**Authors:** Yusuf Karadeniz, Fatma Özpamuk-Karadeniz, Süleyman Ahbab, Esra Ataoğlu, Günay Can

**Affiliations:** 1Division of Endocrinology and Metabolism, Department of Internal Medicine, Faculty of Medicine, Necmettin Erbakan University, Konya 42010, Turkey; dryusufkaradeniz@gmail.com; 2Department of Cardiology, Private Büyükşehir Hospital, Konya 42060, Turkey; 3Department of Internal Medicine, University of Health Sciences, Haseki Training and Research Hospital, Istanbul 34270, Turkey; drsahbab@gmail.com (S.A.); eataoglu@gmail.com (E.A.); 4Departments of Public Health, Cerrahpaşa Medical Faculty, Istanbul University, Istanbul 34098, Turkey; gunaycan09@yahoo.fr

**Keywords:** vitamin D, blood pressure, hypertension, parathyroid hormone

## Abstract

*Background and objectives*: Hypertension is a global health problem and a major risk factor for cardiovascular diseases. Vitamin D deficiency is closely related to high blood pressure and the development of hypertension. This study investigated the relationship between the vitamin D and blood pressure status in healthy adults, and their 8-year follow-up was added. *Materials and Methods*: A total of 491 healthy middle-aged participants without any chronic illness, ages 21 to 67 at baseline, were divided into two groups as non-optimal blood pressure (NOBP) and optimal blood pressure (OBP). NOBP group was divided into two subgroups: normal (NBP) and high normal blood pressure (HNBP). Serum 25-hydroxy vitamin D levels were measured with the immunoassay method. 8-year follow-up of the participants was added. *Results*: The average vitamin D level was detected 32.53 ± 31.50 nmol/L in the OBP group and 24.41 ± 14.40 nmol/L in the NOBP group, and a statistically significant difference was found (*p* < 0.001). In the subgroup analysis, the mean vitamin D level was detected as 24.69 ± 13.74 and 24.28 ± 14.74 nmol/L in NBP and HNBP, respectively. Together with parathyroid hormone, other metabolic parameters were found to be significantly higher in the NOBP. During a median follow-up of 8 years, higher hypertension development rates were seen in NOBP group (*p* < 0.001). *Conclusions*: The low levels of vitamin D were significantly associated with NBP and HNBP. The low levels of vitamin D were also associated with the development of hypertension in an 8-year follow-up.

## 1. Introduction

Vitamin D deficiency is a very common health problem worldwide, and studies have shown that vitamin D deficiency is common in the Turkish population [1,2,3]. In addition to the well-known function of calcium and bone metabolism, vitamin D deficiency is related to many chronic diseases, such as metabolic syndrome, hypertension, diabetes mellitus, coronary artery disease, and heart failure [4]. Besides cardiovascular diseases, it has a role in inflammatory, autoimmune, neurodynamic diseases, and cancer etiology [5]. Except from systemic diseases, it has been shown that decreased vitamin D and increased cytokine levels in saliva are associated with local diseases, such as periodontal disease [6].

Hypertension is the main preventable cause of premature death and disability in the world [7]. Hypertension has many etiological factors, which include age, race, family history, obesity, sedentary lifestyle, using tobacco, high salt intake, stress, and consuming alcohol in a larger quantity. It is thought that vitamin D deficiency is one of these etiological reasons, and vitamin D level is inversely related to blood pressure (BP) and incident hypertension. However, the blood pressure-lowering effect of vitamin D replacement was not observed in most studies; it was found to be effective in a few studies [8]. Various studies revealing the mechanism between vitamin D and hypertension have been presented, and studies are continuing on this subject [9].

Pre-hypertension is the pre-clinic stage of hypertension and is an important risk factor for both the development of hypertension and cardiovascular diseases [10]. According to the latest ESC guideline, pre-hypertension was defined as normal and high normal instead [11]. The onset of hypertension can be delayed or prevented with adequate primary prevention. Most of the studies examined the association between vitamin D and BP but did not focus on pre-hypertension or the development of hypertension in the pre-hypertensive, low vitamin level group on long-term follow-up. To the best of our knowledge, no study has been conducted on the relationship between vitamin D pre-hypertension association and future hypertension development risk in the Turkish population. This study aimed to investigate the association between vitamin D status and pre-hypertension in healthy, disease-free adults. In addition, the 8-year follow-up of these patients was added, and the rate of hypertension and other chronic events were investigated.

## 2. Materials and Methods

### 2.1. Participants

This study consisted of 155 males and 336 females, a total of 491 individuals who were admitted to our hospital’s outpatient clinic between December 2012 and July 2013 for routine health controls. The patients were followed for 8 years after admission. Physical examination findings and anthropometric measurements were noted. Biochemical parameters and plasma 25-hydroxy vitamin D (25-OH vitamin D) levels were analyzed. Participants were divided into two groups, normal to high normal blood pressure (NBP-HNBP) and optimal blood pressure (OBP), according to 2018 ESC/ESH Clinical Practice Guidelines for hypertension [11]. We included NBP and HNBP in the non-optimal blood pressure group (NOBP). There were 289 people with NOBP, 101 men and 188 women, and 202 people with OBP, 54 of whom were male, and 148 were female. Patients diagnosed and treated before with a chronic disease, such as diabetes mellitus, hypertension, coronary artery disease, chronic renal, and liver disease; with an active infection; smokers; and who had received vitamin D treatment in the last 6 months were excluded. Data from the 8-year follow-up period were obtained from medical records. This study was approved by the hospital’s ethics committee. Informed consent was obtained from all participants.

### 2.2. Measurements

BP was measured twice with an aneroid sphygmomanometer from the right arm of all participants in a sitting position after 5 min of rest. BP values were reported in millimeters of mercury (mmHg). Hypertension is defined as office systolic BP values ≥ 140 mmHg and/or diastolic BP values ≥ 90 mmHg according to 2018 ESC guidelines. We excluded those with hypertension in the current assessment. In the ESC guidelines, BP was defined as optimal blood pressure (OBP) < 120–80 mmHg; normal blood pressure (NBP) 120–129 and/or 80–84 mmHg; high normal blood pressure (HNBP) 130–139 and/or 85–89 mmHg according to office BP. The development of hypertension was defined by the evidence of systolic BP ≥ 140 mmHg, and/or diastolic BP ≥ 90 mmHg, and/or current antihypertensive therapy [11].

Weight was measured with light clothing and without shoes. A standard tape measure was used to determine the height and waist circumference, which was measured between the lowest rib and crista iliaca superior. Body mass index (BMI) was computed from values of weight/height squared and universally expressed in kg/m^2^, resulting from mass in kilograms and height in meters.

### 2.3. Laboratory Analysis and Statistics

The whole blood samples were taken after overnight fasting and analyzed in a central laboratory. All laboratory determinations were analyzed with Beckman Coulter AU 2700 auto-analyzer. Plasma 25-OH vitamin D was measured using the DiasorinLiason device by immunoassay method. Vitamin D levels were assessed according to the European Society of Endocrinology guideline. Glucose, creatinine, uric acid, Ca, P, albumin, GGT, LDH, ALP, AST, ALT, triglyceride, total cholesterol, HDL, and LDL levels were measured with Beckman Coulter AU 2700 auto analyzer with the photometric method. Intact PTH, insulin, fT3, fT4, and TSH levels were measured with Beckman Coulter Unicell DXI 800 device with immunoassay method. HsCRP level was measured with Coulter AU 680 device with the turbidimetric method. C-peptide levels were measured with the immunoassay method on the DiasorinLiason device. HbA1c levels were measured with the boronate affinity-based HPLC method on the Premiere Hb 9210 device. Insulin resistance was expressed as The Homeostasis Model Assessment of Insulin Resistance (HOMA-IR) and calculated as (fasting insulin (mIU/mL) × fasting glucose (mg/dL)/405).

Statistical analysis was performed by using Statistical Package for Social Sciences (SPSS) for Windows 16.0. Numerical values were expressed as main ± standard deviation. Statistical associations were assessed using the Student *t*-test (parametric test) or Mann–Whitney U test (non-parametric test). Logistic regression modeling was performed to identify a relationship between NOBP, 25-OH vitamin D, and HOMA-IR. A *p*-value < 0.05 was statistically significant.

## 3. Results

A total of 491 patients were divided into two groups: NOBP and OBP. NOBP group included two subgroups: NBP and HNBP. A total of 289 people with NOBP, 101 men and 188 women, and 202 people with OBP, 54 of whom were male, and 148 were female, were analyzed.

### 3.1. Associations of 25-OH Vitamin D and Normal-High Normal Blood Pressure

A significantly lower mean vitamin D level of 27.8 nmol/L was detected in all patient populations. The mean levels of vitamin D were found to be 24.41 ± 14.40 nmol/L in NOBP group and 32.53 ± 31.50 nmol/L in OBP, and there was a significant statistical difference found (*p* < 0.001) (Table 1). Figure 1 shows the distribution in the whole group according to the vitamin D level that is divided into groups with intervals of 5 nmol/L. In the Figure 1, most of the patients had low vitamin D levels less than <50 nmol/L (Figure 1).

Figure 2 shows the levels of vitamin D levels by dividing into 25 nmol/L quartiles into the study groups (NOBP and OBP). While the number of patients in the first quartile with the lowest vitamin D levels (<25 nmol/L) was higher, the number of patients in the fourth quartile with the higher vitamin D levels (>75 nmol/L) were lower in both study groups, and a serious deficiency of vitamin D < 25 nmol/L were detected in 63.3% of the patients in the NOBP group (Figure 2).

Vitamin D and parathyroid hormone (PTH) levels between groups according to gender were shown in Figure 3. In both groups, vitamin D levels were higher in men compared to women. In contrast to vitamin D, PTH levels were higher in women compared to men (Figure 3).

The characteristics of participants according to 25-OH vitamin D and PTH levels and quartiles of BP levels were shown in Figure 4. Vitamin D average gradually decreases from the OBP group to HNBP group, and the difference between the groups was statistically significant (*p* = 0.008). The mean PTH gradually increases from the OBP group to HNBP group, but the difference between the groups was not statistically significant (*p* = 0.119) (Figure 4).

The mean systolic BP at baseline was 130.53 ± 6.53 and 107.65 ± 6.61, and diastolic BP was81.73 ± 6.90 and 67.53 ± 6.19 in NOBP and OBP group, respectively, and there was a statistically significant difference between the two subgroups (*p* < 0.001) (Table 1).

### 3.2. Associations of Other Metabolic Parameters and Non-Optimal Blood Pressure

Table 1 shows the baseline characteristics of the study population. The average age is 34.32 ± 8.50 in OBP group and 37.81 ± 9.87 in NOBP group. Those in the NOBP group were more obese and had higher levels of total cholesterol, LDL, triglyceride, uric acid, AST, ALT, CRP, glucose, HbA1c, insulin, C-peptide, PTH, and HOMA-IR levels compared to OBP group (*p* < 0.001). Creatinine, calcium, phosphor, albumin, TSH, fT4, and fT3 levels did not differ between the two groups (Table 1).

In the logistic regression model, when the effect of the specified variables on NOBP are examined, the risk of developing NOBP increases with age (*p* = 0.047, OR = 1.024), 25-OH vitamin D (*p* = 0.021, OR = 0.985), HOMA-IR (*p* = 0.047, OR = 1.229), and BMI (*p* = 0.000, OR = 1.199) (Table 2).

### 3.3. Development of Incident Hypertension and Other Chronic Diseases during Follow-Up

Among 491 participants free of hypertension, 79 of them developed incident hypertension during an 8-year follow-up. There were 61 patients in the NOBP group and 18 patients in the OBP group (Table 3).

The rate of developing hypertension was found to be significantly higher in the NOBP group with low vitamin D levels (*p* < 0.001) (Table 3). The development of coronary artery disease in OBP was 5.9% and 7.6% in the NOBP group, and there was no statistical significance between the two groups (*p* = 0.46). The development of diabetes mellitus in the OBP was 8.0% and 13.5% in the NOBP group, and the NOBP group was found to be more susceptible to diabetes (*p* = 0.07) (Figure 5). The odds ratio for the development of HT in NOBP was 2.67.

## 4. Discussion

The association between vitamin D levels and BP was previously reported in several numbers of studies, and lower 25-OH vitamin D levels were associated with higher blood pressures and higher prevalence of hypertension [12,13,14,15]. However, there is no consensus on this issue, and there are studies that do not support this association. Unlike other studies, our study specifically examined the relationship between vitamin D and normal-high normal pressure in disease-free adults and examined the development of hypertension and other events in long-term follow-up. There was a strong negative association between vitamin D and BP levels in our study population. Like our study, Jorde et al. revealed a strong relationship between hypertension and vitamin D. However, there was no association between vitamin D and future hypertension development different from our study [16]. In the study by Sabanayagam et al., low vitamin D levels were found to be associated with pre-hypertension in U.S. adults [17]. Although Turkey has high sunlight exposure, vitamin D deficiency is still seen at high rates, and baseline vitamin D levels were very low in the study population due to the ethnic structure of the Turkish society. In the meta-analysis, although 63% of vitamin D deficiency was detected in the Turkish population, this rate was found to be higher in our study group [18]. The reason for this can be that they live in a metropolis like Istanbul, and as a result, they may spend long periods in closed areas. This percent is found in approximately 40% of the European population and is lower than in the Turkish population [19].

It has been shown that up-regulation of the renin-angiotensin-aldosterone system (RAAS) is an important risk factor for the development of hypertension [20,21,22]. Vitamin D has been shown to affect the RAAS in various ways by binding to vitamin D receptors in different human and animal studies [23,24]. Forman et al. showed that lower 25-OH vitamin D levels were associated with higher circulating angiotensin 2 levels and blunted renal plasma flow responses to infused angiotensin II in renal plasma flow in vitamin D insufficient group compared with sufficient vitamin D levels [23]. This increase in angiotensin 2 levels leads to hypertension, cardiac hypertrophy, and increased water intake. The suppression of renin expression by vitamin D is different from its role in calcium metabolism. This is associated with the sensing mechanism of the volume with salt and angiotensin 2 feedback regulations [25]. Li et al. found that renin expression was increased in mice whose vitamin D receptor was removed. As a result of increased renin level, angiotensin II increased and lead to the development of hypertension and cardiac hypertrophy in vitamin D receptor null mice [22]. They also showed that 25-OH vitamin D directly inhibits renin gene transcription [26]. Vaidya et al. found that the polymorphism in the Fok1 vitamin D receptor gene caused an increase in plasma renin level and the development of hypertension. The results showed that vitamin D is a potential regulator of renin activity in humans [27]. These results showed that vitamin D deficiency increased RAAS activation.

Secondarily, other than the RAAS, low vitamin D levels are associated with increased insulin resistance, which plays a role in the pathogenesis of hypertension [28].

Third, the study shows that vitamin D inhibits vascular muscle cell proliferation; as a result, low levels of vitamin D cause hypertension development as a result of vascular muscle cell proliferation [29].

Vitamin D replacement has been found to have an antihypertensive effect in the reported studies [30]. On the contrary, in some studies, the antihypertensive effect of vitamin D replacement could not be demonstrated, and it has not been revealed at what rate the development of hypertension could be prevented or delayed if vitamin D replacement was given to this pre-hypertensive group. In the study, vitamin D3 treatment in obese hypertensive’s regulates angiotensin 2 activity in tissues and increases angiotensin 2 sensitivity in tissues similar to angiotensin converting enzyme inhibitors [31]. The effect of vitamin D replacement on pre-hypertension and stage 1 hypertension was not shown in the DAYLIGHT randomized, prospective study conducted with 591 patients [32]. The effectiveness of vitamin D for the treatment of hypertension can be due to characteristics of the study population, different sample sizes, short duration of treatment, low-dose usage of vitamin D, and lower follow-up period. Further studies with a bigger study population must be done.

In addition to its well-known role in calcium metabolism, high PTH levels have been shown to increase many cardiovascular risk factors along with hypertension [33,34,35,36]. Both secondary hyperparathyroidism and hypocalcemia due to vitamin D deficiency and primary hyperparathyroidism independent from vitamin D levels are associated with cardiovascular pathology. In a meta-analysis, circulating PTH levels were associated with a higher risk of arterial hypertension. In an American sample with 3002 participants, higher serum PTH levels were significantly associated with a greater risk of incident hypertension [37]. We assessed simultaneously both vitamin D and PTH, and our findings showed that higher PTH levels were significantly associated with higher BP levels. BP increasing the effect of PTH except in regulating the 25-OH vitamin D level can be due to increasing renin release from the kidney, causing endothelial dysfunction, increasing arterial stiffness, and activating the sympathetic system [38,39,40].

Vitamin deficiency has also been shown to be a risk factor for metabolic syndrome. An inverse relationship was found between vitamin D levels and increased BP, high triglycerides, low HDL, increased waist circumference, and elevated fasting glucose, which are components of metabolic syndrome [41]. HNBP is also one of the components of metabolic syndrome, and a strong inverse relationship was found with vitamin D in our study with other metabolic syndrome components.

Classically, serum 25 OH form of vitamin D is measured to show the level of vitamin D. We measured the 25-OH vitamin D with the immunoassay method. Slominski et al. detected CYP11A1-derived secosteroids in the human epidermis and serum as well as the porcine adrenal gland, which acted as hormones in vivo [42]. They detected these active metabolites of vitamin D by using liquid chromatography-tandem mass spectrometry [43]. They also revealed that CYP11A1 plays an important role in the skin’s protective barrier and immune functions by playing a role in local steroidogenesis and vitamin D metabolism in the skin [44]. From these findings, although secosteroids have a role in the cellular effects of vitamin D, circulating 25-hydroxycholecalciferol is considered a clinical indicator of vitamin D status [45].

As expected, we have investigated that the development of hypertension was extremely higher in the NOBP subjects compared with the OBP subjects in an 8-year follow-up. As opposed to hypertension, coronary artery disease development was not found high in the NOBP group. The development of diabetes tended to be higher in the NOBP group compared to the OBP group.

The strengths of our study are that the significance of the results is very high. In addition, by using vitamin D together with PTH and other laboratory parameters, the independent prognostic value of these values was determined. In addition, our study has a long follow-up period of eight years. Numerous studies and meta-analyses have suggested that vitamin D deficiency has a negative association with hypertension; however, this effect has been seldom studied in the Turkish population.

The limitations of our study are a low number of patients, and they are from a single center. The other limitation is population-based nature. The seasonal variation of vitamin D may also affect its levels. Although Liquid Chromatography-tandem Mass Spectrometry method is the gold standard method for vitamin D measurement now, we used the immunoassay method. Another possible limitation is the measurement of vitamin D and hypertension at a single fixed time. Although 1.25-OH vitamin D is the active metabolite, only 25-OH vitamin D could be measured in this study. Dietary habits or the use of salt could not be standardized for the development of hypertension.

## 5. Conclusions

In conclusion, our study suggests that serum 25-OH vitamin D levels are independently associated with normal to high normal blood pressure and risk of incident hypertension in a middle-aged Turkish population in the prospective study. Along with the negative correlation between vitamin D level and blood pressure in non-hypertensive individuals, the development of hypertension in patients with high-normal and normal blood pressure in an 8-year follow-up is a striking result. Further population studies with a large number of patients are needed to evaluate the role of serum 25-OH vitamin D supplementation and/or level in preventing or delaying the development of hypertension in pre-hypertensive patients. Since vitamin D values in our study group and Turkish society were found to be very low, a community nutrition program should be developed to increase its level.

## Figures and Tables

**Figure 1 medicina-57-01297-f001:**
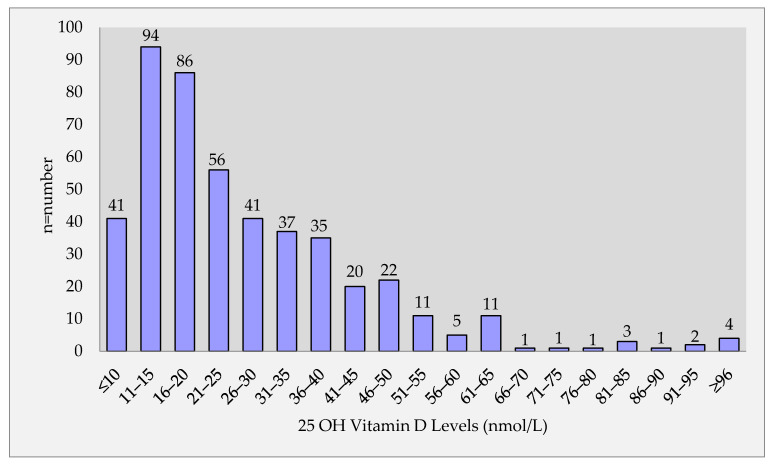
Distribution of patients according to vitamin D levels.

**Figure 2 medicina-57-01297-f002:**
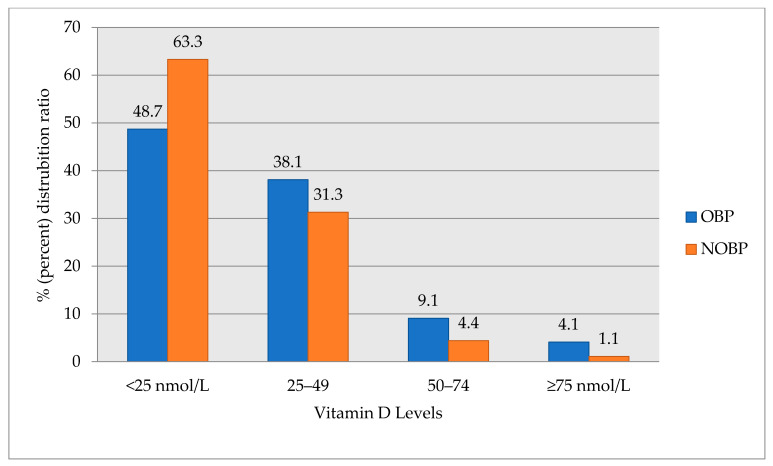
Distribution of vitamin D levels by groups. OBP, optimal blood pressure; NOBP, non-optimal blood pressure.

**Figure 3 medicina-57-01297-f003:**
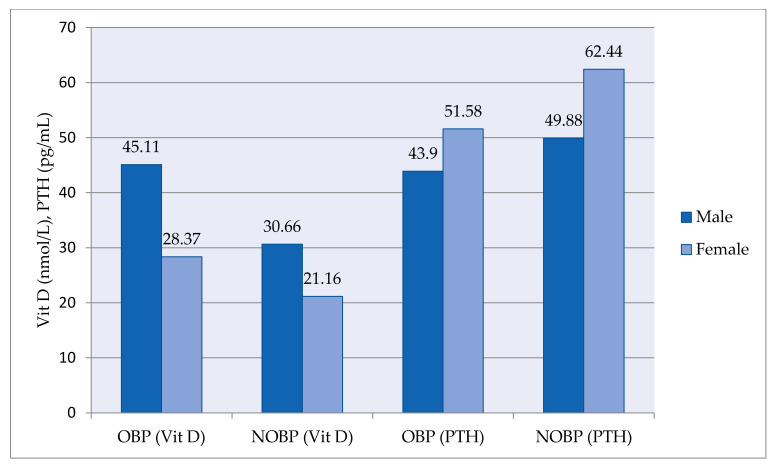
Comparison of vitamin D and PTH levels between groups (OBP and NOBP) by gender. OBP, optimal blood pressure; NOBP, non-optimal blood pressure; VitD, vitamin D; PTH, parathyroid hormone.

**Figure 4 medicina-57-01297-f004:**
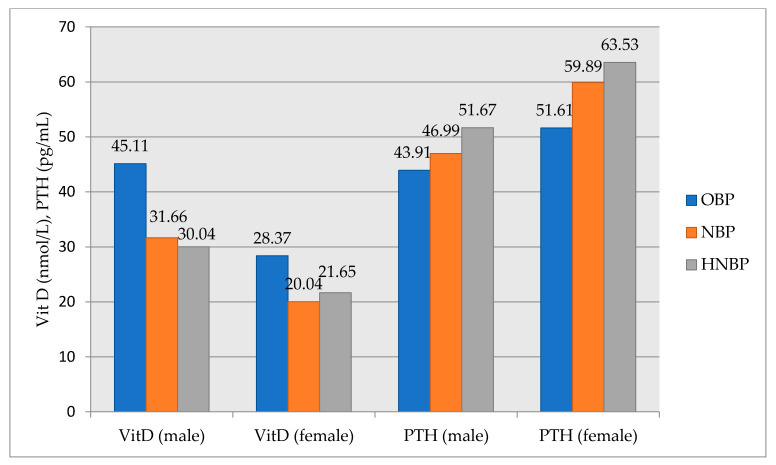
Comparison of vitamin D and PTH levels between subgroups (OBP, NBP, and HNBP) by gender. OBP, optimal blood pressure; NBP, normal blood pressure; HNBP, high-normal blood pressure; VitD, vitamin D; PTH, parathyroid hormone.

**Figure 5 medicina-57-01297-f005:**
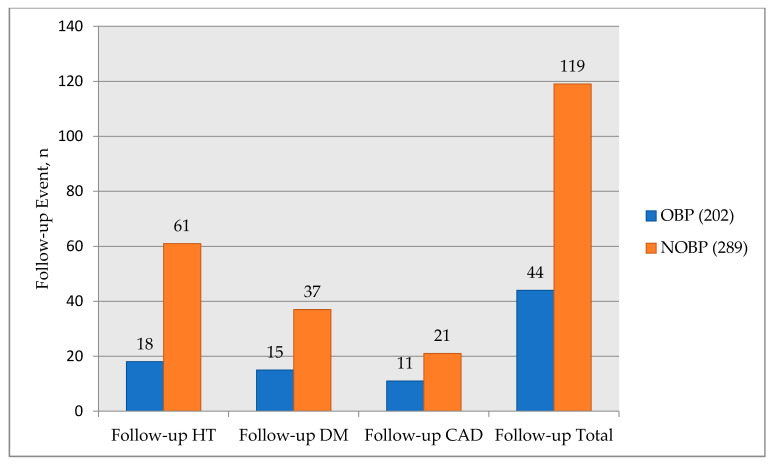
Comparison of events at 8-year follow-up. OBP, optimal blood pressure; NOBP, non-optimal blood pressure; *n*, number; HT, hypertension; DM, diabetes mellitus; CAD, coronary artery disease.

**Table 1 medicina-57-01297-t001:** Comparison of anthropometric, blood pressure, and metabolic parameters between NOBP and OBP groups.

	OBP (*n* = 202)		NOBP (*n* = 289)		
	Mean	SS	Mean	SS	*p*
Age, years	34.32	8.50	37.81	9.87	<0.001
SBP, mm Hg	107.65	6.61	130.53	6.53	<0.001
DBP, mmHg	67.53	6.19	81.73	5.90	<0.001
MABP, mmHg	80.90	5.74	98.00	5.53	<0.001
Height, cm	164.90	8.54	164.90	9.74	= 0.998
Weight, kg	68.32	12.41	80.69	14.43	<0.001
Body mass index, kg/m^2^	25.10	4.50	29.73	5.14	<0.001
Waist circumference, cm	83.82	10.26	95.57	11.06	<0.001
Fasted glucose, mg/dL	91.06	11.04	95.23	11.84	<0.001
Creatinine, mg/dL	0.67	0.15	0.66	0.14	0.871
LDL, mg/dL	115.23	31.24	122.85	36.39	= 0.017
Triglyceride, mg/dL	99.04	55.84	151.23	133.01	<0.001
Total cholesterol, mg/dL	184.73	37.15	197.59	44.39	<0.001
Uric acid, mg/dL	4.06	0.99	4.70	1.30	<0.001
AST, U/L	20.24	6.05	23.50	11.25	<0.001
ALT, U/L	18.77	10.39	26.45	22.83	<0.001
GGT, U/L	19.45	14.70	29.26	26.31	<0.001
ALP, U/L	72.92	24.28	81.48	25.24	<0.001
Albumin, g/L	4.36	0.31	4.32	0.29	= 0.216
Calcium, mg/dL	9.57	0.50	9.60	0.48	= 0.518
Phosphor, mg/dL	3.42	0.56	3.35	0.59	= 0.258
PTH, pg/mL	49.60	23.98	58.08	26.33	<0.001
Vitamin D (25-hydroxyvitamin D), nmol/L	32.53	31.50	24.41	14.40	<0.001
HbA1c, %	5.33	0.46	5.52	0.39	<0.001
Fasted insulin, mU/L	6.96	7.13	9.58	7.61	<0.001
HOMA-IR	1.60	1.74	2.32	2.08	<0.001
C peptide, ng/mL	2.32	0.92	3.16	1.68	<0.001
FT3, ng/dL	3.23	0.38	3.22	0.40	= 0.246
FT4, ng/dL	0.84	0.13	0.82	0.15	= 0.330
TSH, mIU/L	1.81	1.31	1.95	1.32	= 0.238
CRP, mg/L	0.96	2.17	2.08	4.05	<0.001

SBP, systolic blood pressure; DBP, diastolic blood pressure; MABP, mean arterial blood pressure; LDL, low-density lipoprotein; AST, aspartate aminotransferase; ALT, alanine aminotransferase; GGT, gamma-glutamyl transferase; ALP, alkaline phosphatase; HOMA-IR, Homeostatic Model Assessment for Insulin Resistance; FT3, free triiodothyronine; FT4, free thyroxine; TSH, thyroid-stimulating hormone; CRP, C-reactive protein; PTH, parathyroid hormone.

**Table 2 medicina-57-01297-t002:** The multivariate logistic regression model.

	*p*	OR	%95 Confidence Interval	
			Low	High
Age	0.047	1.024	1.000	1.048
25 (OH)vitamin D	0.021	0.985	0.972	0.998
HOMA-IR	0.047	1.229	1.002	1.507
BMI	0.000	1.199	1.133	1.270
Constant	0.000	0.005		

HOMA-IR, Homeostatic Model Assessment for Insulin Resistance; BMI, body mass index.

**Table 3 medicina-57-01297-t003:** Development of Hypertension, diabetes mellitus, and coronary artery disease at 8-year follow-up.

	OBP (*n* = 202)	NOBP (*n* = 289)	*p*-Value
Development of HT	18	61	<0.001
Development of DM	15	37	=0.07
Development of CAD	11	21	=0.46

OBP, optimal blood pressure; NOBP, non-optimal blood pressure; n, number; HT, hypertension; DM, diabetes mellitus; CAD, coronary artery disease.

## Data Availability

The datasets generated for this study are available on request to the corresponding author.

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
