# Peer review of "Vitamin D Deficiency Is a Potential Risk for Blood Pressure Elevation and the Development of Hypertension"

_medicina, 2021, doi:10.3390/medicina57121297_

Round 1
Reviewer 1 Report
this paper remains fundamentally flawed. vitamin D is not related to blood pressure control. See for instance metanalysis by Barbarawi et al (JAMA cardiol 2019). Bolland et al (Lancet diab endocrinol 2014) in his 'futility analysis' paper showed that by 2006 enough clinical trials had been perfomed of vit D supplementation to absolutely confirm the absence of effect. this paper adds nothing and will merely continue the confusion around Vit D correlation (BUT NOT CAUSATION) and BP
Author Response
Dear Editor-in-Chief,
Appended I am submitting the manuscript titled ‘Vitamin D deficiency is a potential risk for blood pressure elevation and the development of hypertension’ a prospective cross-sectional study to be considered for publication in Medicina.
The manuscript has not been previously published or accepted for publication and is not submitted or under simultaneous review for publication. We did all the comments that reviewers suggested in track change format. It complies with ethical considerations. All authors have participated in the study and approved the manuscript. We state that no potential conflict of interest of authors exists in regard to the study. I am looking forward to hear from you on the result of the manuscript.
With best regards,
Fatma Özpamuk-Karadeniz, MD
Response to Reviewer 1 Comments
Point 1: this paper remains fundamentally flawed. vitamin D is not related to blood pressure control. See for instance metanalysis by Barbarawi et al (JAMA cardiol 2019). Bolland et al (Lancet diab endocrinol 2014) in his 'futility analysis' paper showed that by 2006 enough clinical trials had been perfomed of vit D supplementation to absolutely confirm the absence of effect. this paper adds nothing and will merely continue the confusion around Vit D correlation (BUT NOT CAUSATION) and BP
Response for Point 1: As we mentioned in the text, since there are studies with opposing views on the vitamin D-hypertension relationship, it is clear that new studies are needed. Our study strongly reveals this negative relationship. The negative correlation between vitamin d level and blood pressure in non-hypertensive individuals and the development of hypertension in patients with high-normal and normal blood pressure in 8-year follow-up are striking results of our article. Of course, large-scale studies are needed in this direction. We hope that our article will be a pioneer to shed light on larger studies. Regards.

Reviewer 2 Report
i do not have any comments.
Author Response
Dear Editor-in-Chief,
Appended I am submitting the manuscript titled ‘Vitamin D deficiency is a potential risk for blood pressure elevation and the development of hypertension’ a prospective cross-sectional study to be considered for publication in Medicina.
The manuscript has not been previously published or accepted for publication and is not submitted or under simultaneous review for publication. We did all the comments that reviewers suggested in track change format. It complies with ethical considerations. All authors have participated in the study and approved the manuscript. We state that no potential conflict of interest of authors exists in regard to the study. I am looking forward to hear from you on the result of the manuscript.
With best regards,
Fatma Özpamuk-Karadeniz, MD
Response to Reviewer 2 Comments
Point 1: I do not have any comments.
Response for Point 1: Thank you for your efforts and well-opinion.

Reviewer 3 Report
No further comments.
Author Response
Dear Editor-in-Chief,
Appended I am submitting the manuscript titled ‘Vitamin D deficiency is a potential risk for blood pressure elevation and the development of hypertension’ a prospective cross-sectional study to be considered for publication in Medicina.
The manuscript has not been previously published or accepted for publication and is not submitted or under simultaneous review for publication. We did all the comments that reviewers suggested in track change format. It complies with ethical considerations. All authors have participated in the study and approved the manuscript. We state that no potential conflict of interest of authors exists in regard to the study. I am looking forward to hear from you on the result of the manuscript.
With best regards,
Fatma Özpamuk-Karadeniz, MD
Response to Reviewer 3 Comments
Point 1: No further comments.
Response for Point 1: Thank you for your efforts and well-opinion.

Reviewer 4 Report
The article entitled”Vitamin D deficiency is a potential risk for blood pressure elevation and the development of hypertension” investigated the relationship between the vitamin D and blood pressure status in healthy adults. Authors have well revised several issues; however, I ask authors to add some key concepts
Authors must discuss more on the correlation regarding oral pathologies that can influence the systemic condition of individuals with low serum levels of vitamin D in order to stress the importance of this parameter in each body area (please, see doi: 10.3390 / ijms21082669)
Minor issues:
Conclusions cannot be reduced to a sentence: you must improve them highlighting the limits and the future insights pointed out from this article.
Several moderate typos are present in the text, please, amend.
Author Response
Dear Editor-in-Chief,
Appended I am submitting the manuscript titled ‘Vitamin D deficiency is a potential risk for blood pressure elevation and the development of hypertension’ a prospective cross-sectional study to be considered for publication in Medicina.
The manuscript has not been previously published or accepted for publication and is not submitted or under simultaneous review for publication. We did all the comments that reviewers suggested in track change format. It complies with ethical considerations. All authors have participated in the study and approved the manuscript. We state that no potential conflict of interest of authors exists in regard to the study. I am looking forward to hear from you on the result of the manuscript.
With best regards,
Fatma Özpamuk-Karadeniz, MD
Response to Reviewer 4 Comments
The article entitled”Vitamin D deficiency is a potential risk for blood pressure elevation and the development of hypertension” investigated the relationship between the vitamin D and blood pressure status in healthy adults. Authors have well revised several issues; however, I ask authors to add some key concepts
Point 1: Authors must discuss more on the correlation regarding oral pathologies that can influence the systemic condition of individuals with low serum levels of vitamin D in order to stress the importance of this parameter in each body area (please, see doi: 10.3390 / ijms21082669)
Response for Point 1: We added a sentence as you suggested …. Except from systemic diseases, it has been shown that decreased vitamin D and increased cytokine levels in saliva are associated with local diseases such as periodontal disease [6]. ….
- Costantini, E.; Sinjari, B.; Piscopo, F.; Porreca, A.; Reale, M.; Caputi, S.; Murmura, G. Evaluation of Salivary Cytokines and Vitamin D Levels in Periodontopathic Patients. Int. J. Mol. Sci. 2020, 21, 2669.
Minor issues:
Point 2: Conclusions cannot be reduced to a sentence: you must improve them highlighting the limits and the future insights pointed out from this article.
Response for Point 2: We improved this part as you suggested.
Point 3: Several moderate typos are present in the text, please, amend.
Response for Point 3: We checked the manuscript again and changed the typos as you suggested.

This manuscript is a resubmission of an earlier submission. The following is a list of the peer review reports and author responses from that submission.
Round 1
Reviewer 1 Report
This paper remains fundamentally flawed and will only serve to continue the confusion around this area. Despite what the authors say it is quite clear from numerous meta analyses of interventional studies that vitamin D plays no part whatsoever in blood pressure control. The association between low vitamin D levels and higher blood pressure have been extensively recorded and equally extensive interventional studies have shown that there is no causal link. There is no benefit to repeatedly going over this ground. Vitamin D deficiency is conclusively not a risk factor for blood pressure elevation.
Author Response
Dear Editor-in-Chief,
Appended I am submitting the manuscript titled ‘Vitamin D deficiency is a potential risk for blood pressure elevation and the development of hypertension’ a prospective cross-sectional study to be considered for publication in Medicina.
The manuscript has not been previously published or accepted for publication and is not submitted or under simultaneous review for publication. We did all the comments that reviewers suggested. It complies with ethical considerations. All authors have participated in the study and approved the manuscript. We state that no potential conflict of interest of authors exists in regard to the study. I am looking forward to hear from you on the result of the manuscript.
With best regards,
Fatma Özpamuk-Karadeniz, MD
Response to Reviewer 1 Comments
Point 1: This paper remains fundamentally flawed and will only serve to continue the confusion around this area. Despite what the authors say it is quite clear from numerous meta analyses of interventional studies that vitamin D plays no part whatsoever in blood pressure control. The association between low vitamin D levels and higher blood pressure have been extensively recorded and equally extensive interventional studies have shown that there is no causal link. There is no benefit to repeatedly going over this ground. Vitamin D deficiency is conclusively not a risk factor for blood pressure elevation.
Response for Point 1: There are many studies showing the relationship between low vitamin D level and hypertension, but there are some differences in our study and to summarize them; Almost all studies have been designed on the relationship between vitamin D and hypertension. In our study, when we compared non-hypertensive patients according to blood pressure, rather than the relationship between vitamin D and hypertension, a statistically strong relationship was found between vitamin D level, blood pressure level and many metabolic parameters. When these patients were evaluated after a long period of 8 years, it was observed that those with non-optimal blood pressure became hypertensive at a much higher rate than those with optimal blood pressure. With the inference from this study, vitamin D deficiency, which contributes to science, is still at a pandemic level in our study, as it was in the past, and provides up-to-date data in this sense. In this sense, we care about public health because of the predictive nature of our work, and in this sense, we present our work to your appreciation. After your first assessment, we did some changes to improve this article.

Reviewer 2 Report
Dear Editor,
i have reviewed a paper entitled "Vitamin D deficiency is a potential risk for blood pressure ele-vation and the development of hypertension by Yusuf Karadeniz et al.
The manuscript is good wrote and full i regard literature references.
I not have a comments.
Author Response
Dear Editor-in-Chief,
Appended I am submitting the manuscript titled ‘Vitamin D deficiency is a potential risk for blood pressure elevation and the development of hypertension’ a prospective cross-sectional study to be considered for publication in Medicina.
The manuscript has not been previously published or accepted for publication and is not submitted or under simultaneous review for publication. We did all the comments that reviewers suggested. It complies with ethical considerations. All authors have participated in the study and approved the manuscript. We state that no potential conflict of interest of authors exists in regard to the study. I am looking forward to hear from you on the result of the manuscript.
With best regards,
Fatma Özpamuk-Karadeniz, MD
Response to Reviewer 2 Comments
Point 1:
Dear Editor,
I have reviewed a paper entitled "Vitamin D deficiency is a potential risk for blood pressure ele-vation and the development of hypertension by Yusuf Karadeniz et al.
The manuscript is good wrote and full i regard literature references.
I not have a comments.
Response for Point 1: Thank you for your efforts and well-opinion.

Reviewer 3 Report
The article “Vitamin D deficiency is a potential risk for blood pressure elevation and the development of hypertension” has focused on relevant health problems – vitamin D status and its association with the risk of hypertension. The decreased vitamin D serum concentration among different population groups is very common. The main results from this study showed that vitamin D level was significantly higher in optimal blood pressure group as well as associated with the development of hypertension in 8 years follow-up. The results of this study are already known and described in the literature.
Issues that can be improved:
- Materials and methods were not described enough. How much blood was taken for the study? In results section many different biochemical parameters appears (Table 1), while in laboratory analysis section no information about them was given.
- The gold standard for vitamin D determination is the Liquid Chromatography-tandem Mass Spectrometry – it could be included in limitations of the study. The limitations are very short and do not reflects all weakness of this study. Moreover in the first sentence of this section Authors wrote “strengths” – isn’t a mistake?
- No strengths of the study was given.
- In first sentence of the discussion Authors mention “several numbers of studies” and gave only one reference number.
- In discussion more information about mechanisms connected with vitamin D deficiency and hypertension could be given.
- Please check the values of vitamin D level (32.53+31.50 nmol/L?) in the abstract.
- English and scientific language should be improved.
- References should be prepared according to Instructions for Authors of the Journal.
Author Response
Dear Editor-in-Chief,
Appended I am submitting the manuscript titled ‘Vitamin D deficiency is a potential risk for blood pressure elevation and the development of hypertension’ a prospective cross-sectional study to be considered for publication in Medicina.
The manuscript has not been previously published or accepted for publication and is not submitted or under simultaneous review for publication. We did all the comments that reviewers suggested. It complies with ethical considerations. All authors have participated in the study and approved the manuscript. We state that no potential conflict of interest of authors exists in regard to the study. I am looking forward to hear from you on the result of the manuscript.
With best regards,
Fatma Özpamuk-Karadeniz, MD
Response to Reviewer 3 Comments
The article “Vitamin D deficiency is a potential risk for blood pressure elevation and the development of hypertension” has focused on relevant health problems – vitamin D status and its association with the risk of hypertension. The decreased vitamin D serum concentration among different population groups is very common. The main results from this study showed that vitamin D level was significantly higher in optimal blood pressure group as well as associated with the development of hypertension in 8 years follow-up. The results of this study are already known and described in the literature.
Issues that can be improved:
Point 1: Materials and methods were not described enough. How much blood was taken for the study? In results section many different biochemical parameters appears (Table 1), while in laboratory analysis section no information about them was given.
Response for Point 1:Laboratory analysis section was detailed in line with your suggestion. The following has been added to this section:
The whole blood samples were taken after overnight fasting and analyzed in a central laboratory. All laboratory determinations were analyzed with Beckman Coulter AU 2700 auto-analyzer. Plasma 25-OH vitamin D was measured using the DiasorinLiason device by immunoassay method. Vitamin D levels were assessed according to the European Society of Endocrinology guideline. Glucose, creatinine, uric acid, Ca, P, albumin, GGT, LDH, ALP, AST, ALT, triglyceride, total cholesterol, HDL, LDL levels were measured with Beckman Coulter AU 2700 auto analyzer with photometric method. Intact PTH, insulin, fT3, fT4, TSH levels were measured with Beckman Coulter Unicell DXI 800 device with immunoassay method. HsCRP level was measured with Coulter AU 680 device with turbidometric method. C-Peptide levels were measured with the immunoassay method on the Diasorin Liason device. HbA1C levels were measured with the boronate affinity based HPLC method on the Premiere Hb 9210 device.
Point 2: The gold standard for vitamin D determination is the Liquid Chromatography-tandem Mass Spectrometry – it could be included in limitations of the study. The limitations are very short and do not reflects all weakness of this study. Moreover in the first sentence of this section Authors wrote “strengths” – isn’t a mistake?
Response for Point 2: We are sorry about this mistake. We enlarged the limitations part as you suggested. The following has been added to limitation part. ‘Although Liquid Chromatography-tandem Mass Spectrometry method is the gold standard method for vitamin D measurement now, we used immunoassay method’.
Point 3: No strengths of the study was given.
Response for Point 3: We added the strengths part as you suggested. The following has been added.’
Strengths of our study are the significance of the results is very high. In addition, by using vitamin D together with PTH and other laboratory parameters the independent prognostic value of these values ​​was determined. In addition, our study has a long follow-up period of 8 years. Numerous studies and meta-analysis have suggested that vitamin D deficiency has a negative association with hypertension however, this effect has few been studied in Turkish population’.
Point 4: In first sentence of the discussion Authors mention “several numbers of studies” and gave only one reference number.
Response for Point 4: We added references number of 12, 13, 14 as you suggested.
Point 5: In discussion more information about mechanisms connected with vitamin D deficiency and hypertension could be given.
Response for Point 5: We added more mechanisms as you suggest. The following has been added.’ This increase in angiotensin 2 levels leads to hypertension, cardiac hypertrophy and increased water intake. The suppression of renin expression by vitamin D is different from its role in calcium metabolism. This is associated with the sensing mechanism of the volume with salt and angiotensin 2 feedback regulations.
Vaidya et al. found that the polymorphism in the Fok1 vitamin D receptor gene caused an increase in plasma renin level and the development of hypertension. The results showed that vitamin D is a potential regulator of renin activity in humans.
Secondarily, except from the renin-angiotensin-aldosterone system, low vitamin D levels are associated with increased insulin resistance, which plays a role in the pathogenesis of hypertension.
Third, the study shows that vitamin D inhibits vascular muscle cell proliferation, as a result low levels of vitamin D cause hypertension development as a result of vascular muscle cell proliferation.
Point 6: Please check the values of vitamin D level (32.53+31.50 nmol/L?) in the abstract.
Response for Point 6: Since the distribution range of vitamin D levels in the OBP group was wide, the standard deviation was found to be high in this group.
Point 7: English and scientific language should be improved.
Response for Point 7: We tried to correct the errors by reviewing the language again.
Point 8: References should be prepared according to Instructions for Authors of the Journal.
Response for Point 8:The references were arranged according to the journal rules, but we reviewed it again.
